5

# A long term study of polar ozone loss derived from data assimilation of Odin/SMR observations

Kazutoshi Sagi<sup>1</sup> and Donal Murtagh<sup>1</sup>

<sup>1</sup>Department of Earth and Space Sciences, Chalmers University of Technology, Gothenburg, Sweden *Correspondence to:* Kazutoshi Sagi sagi@chalmers.se

**Abstract.** Odin, a Swedish-led satellite project in collaboration with Canada, France and Finland, was launched on 20 February 2001 and continues to produce profiles of chemical species relevant to understanding the middle and upper atmosphere. Long-term observations of stratospheric ozone are useful for trend analysis of chemical ozone loss. This study concerns ozone loss over both poles utilizing 12 years of ozone data from Odin/Sub-Millimetre Radiometer (SMR). We have applied the data assimilation technique described by Rösevall et al. (2007) with a number of improvements to study the inter-annual variability during the entire Odin period. The chemical ozone losses at potential temperature levels between 425 K and 950 K

- , ( corresponding to an altitude range of 15 to 40 km approximately 90 hPa and 7 hPa in pressure), are derived.
   Two SMR ozone products retrieved from the emission lines centred at 501 GHz and 544 GHz were used. An internal com-
- parison of the two analyses using 501 GHz and 544 GHz ozone has been carried out by inspecting the vortex mean ozone in
  March and October during 2002 2013 and 2003 2012 in the Northern and Southern Hemisphere, respectively. Ozone derived from data assimilation using the two data sets match within 10% at the levels studied, while below 550 K in the Southern Hemispheremore than 50% of the difference is found. Here, 544 GHz ozone is 0.5 parts per million volume (ppmv) lower than 501 GHz ozone because of better sensitivity in 544 GHz ozone in the lower stratosphere. Comparisons with other studies have been mainly performed against Sonkaew et al. (2013) since Sonkaew et al. (2013) is one of the few studies having consistent
- 15 estimations of ozone depletion using a SCanning Imaging Absorption SpectroMeter for Atmospheric CHartographY (SCIA-MACHY) from 2002 to 2009. 544 GHz ozone loss in the Arctic winter 2004/2005 is in good agreement with SCIAMACHY loss below 450 K to within 0.2 ppmv, while showing no loss around 550 K where SCIAMACHY detected 0.5 ppmv loss. The comparison of Antarctic ozone depletions with Kuttippurath et al. (2015) shows agreement with MLS ozone loss within 0.1 ppmv, while our results were constantly 0.3 ppmv lower than Mimosa-Chim model calculations.
- In the Northern Hemisphere, our assimilation analyses show large inter-annual variability. Three classes of chemical ozone losses are found to occur in cold, warm and intermediate winters between cold and warm. The cold type loss maximises in March below 500 K as in the Southern Hemisphere. The maximum loss in the Northern Hemisphere between 2001/2002 and 2012/2013 was during the cold winter, which happened in 2010/2011 with a loss in volume mixing ratio of 2.1 ppmv at 450 K. Losses of 1.5 ppmv took place at 700 K in the warm winters related to the occurrence of mid-winter major sudden stratospheric
- 25 warming (SSW) events. In the Southern Hemisphere between 2002 and 2012, chemical ozone losses began in mid-August and generally grew to 2.5 ppmv by the end of October. The vertical extent of this loss was 425 – 550 K. All Antarctic winters except

2002 had approximately 80 DU loss in the stratospheric column. In both hemispheres partial columns in the stratosphere show a small increase over the time period from 2002 to 2013, however the statistical confidence is not high enough to identify ozone recovery.

# 1 Introduction

- Ozone depletion and climate change are indirectly linked. Several studies have predicted that the stratospheric cooling induced by the increasing atmospheric carbon dioxide will enhance ozone depletion (Austin et al., 1992; Shindell et al., 1998). In practice, the Arctic lower stratosphere has been getting colder over the past decades (WMO, 2011). The linear dependence, demonstrated by (Rex et al., 2006), between the ozone depletion and the volume of air having temperature below the threshold for polar stratospheric cloud (PSC) formation implies that the stratospheric O<sub>3</sub> depletion in Northern Hemisphere may become
- worse if the cooling trend continues. It is therefore important to have continuous observations and trend analyses of the ozone depletion.

Ozone loss has been quantified by using a variety of techniques based on different assumptions and instruments (e.g. Eichmann et al., 2002; Grooß and Müller, 2003; Rex et al., 2006; Singleton et al., 2007; Tilmes et al., 2004; Tsvetkova et al., 2007). However, most of the studies were done for individual winters in the last decade. For instance, in the Arctic winter

- 2010/2011 several groups reported the unprecedented dramatic ozone depletion over the Arctic polar region approaching that of the Antarctic ozone hole (e.g. Arnone et al., 2012; Manney et al., 2011; Sinnhuber et al., 2011). This winter was obviously different from other Arctic winters from 2000 since the polar vortex was strong and isolated the vortex air from the outside, the polar vortex was sustained by very cold temperatures. For this specific winter we have a number of publications to refer to, while few publications are available for other winters that do not show such dramatic events, i.e unusually cold weather
- in the lower stratosphere or sudden stratospheric warmings (SSW). One long term study of ozone loss was performed using the SCanning Imaging Absorption spectroMeter for Atmospheric CHartographY (SCIAMACHY) (Sonkaew et al., 2013). Sonkaew et al. (2013) estimated ozone loss between 2002 and 2009 and showed a quasi-biennial oscillation (QBO) effect in the inter-annual variability of their derived Arctic ozone losses.
- The Odin satellite was built by Sweden in association with Canada, Finland and France as an observatory aimed at radio astronomy and limb sounding of the Earth's middle atmosphere (Murtagh et al., 2002). It carries two different limb sounding instruments, OSIRIS (Optical Spectrograph/ InfraRed Imaging System) and SMR (Sub-Millimetre Radiometer). Since launch in February 2001, Odin continues to provide data producing a relatively long record of stratospheric ozone. Rösevall et al. (2007, 2008) developed the DIAMOND (Dynamic Isentropic Assimilation Model for OdiN Data) model and estimated polar ozone loss in specific winters such as the Arctic 2002/2003 and Antarctic 2003, Arctic 2004/2005 and Arctic 2006/2007. Sagi et al.
- (2014) updated the DIAMOND model by adding and testing an explicit vertical transport scheme applying it to both Japanese Experiment Module (JEM) / Superconducting Submillimeter-Wave Limb Emission Sounder (SMILES) and Odin/SMR data to study the 2009/2010 Arctic winter. The subject of this paper is to summarize the ozone loss changes on a decadal time-scale by applying the data assimilation technique used in the previous studies to the entire Odin ozone observation period. Generally

previous studies focused on ozone loss below a potential temperature (PT) of 600 K (approximately 24 hPa in pressure and 30 km in altitudes) since the Antarctic ozone holes are mainly caused by chlorine chemistry following PSC formation. However Konopka et al. (2007) showed that, above 600 K ( $\sim$  24 km), the chemical loss induced by the horizontal transport of NO<sub>x</sub> from lower latitudes is as great as the halogen-induced loss below 500 K ( $\sim$  20 km) in the Northern Hemisphere in 2002/2003.

In this paper we have extended the vertical analysis region up to PT of 950 K ( $\sim 40$  km) in order to show the effect of NO<sub>x</sub> transport on ozone losses.

This paper contains the following sections. The methodology and the assimilation model that we used to determine chemical ozone change are described in section 2. Section 3 deals with the SMR instrument, whose stratospheric ozone observations are introduced into the model in this study. Section 4.1 discusses the determination of the polar vortex edge in both hemispheres

during polar winter and spring. Section 4.2 present an internal comparison between ozone losses derived from two different SMR ozone measurements, while section4.3 shows the comparison with other studies. Next we look at the inter-annual variation of ozone loss averaged over the Arctic and Antarctic winters in Sect. 4.4.1 and Sect. 4.4.2, respectively. In section 4.4.3 we have an additional discussion of inter-annual variations in both hemispheres of the lower-stratospheric partial column. Conclusions are presented in the last section.

# 15 2 Methodology

Stratospheric ozone observed by the SMR instrument is affected by not only chemical process but also transport. Thus the unstable nature of the Arctic vortex due to the propagation of planetary waves excited by the complex topography of Northern Hemisphere makes quantifying chemical  $O_3$  loss in the Arctic more difficult. Therefore it is necessary to find a suitable method for extracting the contribution of chemical change in ozone. Using a transport model can help to separate the two processes,

- i.e. transport and chemistry, but we need to ensure that ozone is treated in a consistent manner. Data assimilation is a process by which observations are introduced into a model while constraining these to follow model physics (Lahoz et al., 2010). We have used an updated version of the DIAMOND model (Rösevall et al., 2007) to treat the Odin observations. Two  $O_3$  fields are produced in the model, one is a passive  $O_3$  field that is only transported by advection and another one is an active  $O_3$  field that is modified by assimilation of the Odin/SMR data. The chemical  $O_3$  depletion can be derived by subtracting passive  $O_3$  from
- $25 \quad active \ O_3.$

# 2.1 DIAMOND model

The DIAMOND model is an off-line wind driven isentropic transport and assimilation model designed to simulate horizontal ozone transport in the lower stratosphere with low numerical diffusion (Rösevall et al., 2008). Horizontal off-line wind driven advection has been implemented using the Prather transport scheme (Prather, 1986) which is a mass conservative Eulerian scheme.

The idea of the Prather scheme is that by preserving the zero to second order moments of the sub-grid scale tracer distribution the quality of the transport is preserved. In this study, the wind fields obtained from the European Centre for Medium-Range

Weather Forecasts (ECMWF) operational analyses have been used. Advection calculations are performed on separate layers with constant potential temperature (PT). Since PT in dry air is conservative under adiabatic conditions air parcels normally move on constant PT surfaces. However during polar night conditions considerable descent occurs within the polar vortex and because of the vertical gradients in ozone we must consider the effects of the vertical transport on the estimates of ozone loss.

5 A first-order upstream scheme was implemented in the current version of the model in order to take account of the vertical motion (Sagi et al., 2014). Since the general descent rate inside the vortex is approximately 1–2K per day it is slow enough for the effects of numerical diffusion to be negligible. A vertical range of 425 K–950 K in PT, which corresponds to approximately 18 – 40 km altitude in the polar vortex, was selected for this study.

The tracer profiles from SMR are sequentially assimilated into the advection model. The assimilation scheme in DIAMOND 10 can be described as a variant of the Kalman filter. Details of the assimilation scheme can be found in Rösevall et al. (2008).

### 3 SMR ozone measurements

The Odin satellite was launched into a sun-synchronous dawn-dusk polar orbit. SMR provides vertical profile measurements within the nominal latitude range 82.5°S and 82.5°N at 06:00/18:00 local time at the descending and ascending nodes respectively. In the stratospheric observation mode, two of the receivers, covering the bands centred at 501.8 and 544.6 GHz, are used

- for detecting the spectral emission lines of O<sub>3</sub>, N<sub>2</sub>O, ClO and HNO<sub>3</sub>. Stratospheric observation mode is operated every other day since April 2007 (every third day previous to this). The stratospheric ozone is retrieved from the two different emission lines centred at 501.8 GHz and 544.6 GHz, using the Chalmers version 2.1 and 2.0 retrieval schemes, respectively. Figures 1 and 2 show typical ozone profiles, averaging kernels and errors estimated for the two frequencies for Arctic and Antarctic winters, respectively. The 501 GHz ozone profiles cover the altitude range roughly 17–50 km with an altitude resolution of
- 2–3 km and an estimated single-profile precision of 1.5 parts per million volume (ppmv) (Urban et al., 2005). The filtering criterion used for this study is the measurement response, which is the sum of the rows of the averaging kernel and indicates how much information is derived from the true state in the atmosphere as opposed to coming from the a-priori information. In the analysis, ozone profiles with measurement of response of less than 0.8 have been excluded. It can be seen that 544 GHz ozone measurements show greater sensitivity below 20 km than 501 GHz ozone measurements . This difference can be clearly
- seen in the assimilated results as well. Validation of SMR v2.1 501 GHz ozone data has been performed against balloon sonde measurements as described by Jones et al. (2007). Since there is no validation paper available for SMR v2.0 544 GHz ozone, we show an internal comparison between two ozone data sets retrieved from the two frequencies used in this study. The internal comparison between the two ozone data sets is discussed in section 4.2.