# Peer review of "A long term study of polar ozone loss derived from data assimilation of Odin/SMR observations"

_Atmospheric Chemistry and Physics, 2016_

## Referee Comment (RC1) · Anonymous Referee #1 · 14 Jul 2016

Sagi et al. apply 12-years of Odin/SMR ozone measurements for deriving ozone loss and ozone loss trends in the Northern and Southern Hemisphere. They apply the data assimilation technique described in Rösevall et al. (2007). Internal comparisons as well as comparisons to other data sets have been performed. Overall, the paper presents valuable results, but lacks in writing up of the results. There are a lot of small issues which could have already been resolved before submission. Major revisions are necessary before the paper can be published in ACP.

**General comments:**
1. It is not common to cite other studies in the abstract and it is also absolutely not

necessary. The sentences should be rephrased accordingly (suggestions will be given below). Further, the abstract reads like an introduction and should be thoroughly revised. Especially, the sentences discussing the comparison to other studies should be revised (L12-L19). From the text it does not become clear that these are results of the presented study. Why have these comparisons performed? What do we learn from these comparisons?

2. On several pages throughout the document the line numbering is messed up! Line numbers on one page are increasing and then decreasing again ("10" "15" "5" on page 16) or stuck to one value (e.g. "5" "5" "5" "10" on page 8). The manuscript should be more carefully checked before resubmission. It would probably help to just put the Figures at the end of the manuscript (behind the reference list) as it is the common style for manuscripts.

3. The Sections should be restructured. Section 4 should consist solely of Sub-sections 4.1-4.3 and renamed to e.g. "Vortex mean estimation and data quality assessment". Section 5 would be consist then of three subsections: 5.1 Arctic ozone loss, 5.2 Antarctic ozone loss (would suggest to call it here loss as well rather than hole), and Section 5.3 should be called Arctic and Antarctic Partial Columns. Further, the result should be discussed referring more clearly to the three classes cold, warm and intermediate.

4. A conclusion section should added. What have we learned from this study? How can your results be utilized in future studies?

**Specific comments:**
P1, L5: This is the only reference that may be used in the abstract, but nevertheless the sentence could also e.g. be written as follows without necessarily needing to cite the Rösevall study: "We apply an improved version of the previously used data

assimilation technique, therefore allowing us to study the inter-annual variability during the entire Odin period".

P1, L5: What improvements have been made?

P1, L12: rephrase sentence as follows: "…...in the Southern Hemisphere differences larger than 50
P1, L12: rephrase sentence as follows: "In our study we find that 544 GHz ozone is ……..…"

P1, L14-16: References should not be used in the abstract and the text would be more concise and better readable when shortened a bit. Therefore, I would suggest to rephrase the sentence as follows: "Comparisons with other studies were performed using consistent estimates of ozone depletion from SCIAMACHY (Scanning Imaging Absorption SpectroMeter for Atmospheric CHartografY)." Nevertheless discussing the results of comparisons shown in previous studies are obsolete in the abstract.

P1, L16: It is not clear to what the time period refers to. Is it meant that the comparison was performed for the time period 2002-2009 or that the data was available from 2002-2009?

P1, L18: MLS has not been introduced yet. Further, referencing the Kuttipurath study should be removed from the abstract and the sentence should be rephrased e.g. as follows: The comparison of Antarctic ozone depletion with MLS (Microwave Limb Sounder) shows an agreement in the derived ozone loss within 0.1 ppmv. As mentioned above discussing the results of previous studies in the abstract is obsolete. Thus, it would be better to move these sentences to the introduction.

P1, L21: Why not just name the classes cold, warm and intermediate winters.

Using the expression "Intermediate winters between cold and warm" is quite confusing.

P1, L22: As in the SH? Do you really mean as in the SH or should it just read in the SH?

P1, L20-P2, L2: The results should be presented in a better (more structured) manner. What are the results for each of the three classes and the Northern and Southern Hemisphere, respectively? How large is the column loss for the NH?

P3, L1: The abbreviation PT for potential temperature should be skipped throughout the manuscript.

P3, l2: This holds for both hemispheres and thus the sentence should read "…...Arctic and Antarctic ozone depletion."

P4, L10: To the DIAMOND description a few sentences should be added stating what improvements have been made to the DIAMOND model and that these had been already applied to the 2009/2010 winter in the study by Sagi et al. (2014).

P4, L17ff: The text does not fit to the Figures. Why is for the Arctic only 501 GHz shown and for the Antarctic 544 GHz? There are now assimilated results shown in the Figure. How can you than state (L25) that the assimilated results show the same differences?

P5, Figure 1 caption: "2010/03/05 (YY/MM/DD)". Why so complicated. Why don't you write the date as "25 March 2010" in the caption?

P6, Figure 2 caption: Same here. Why not just writing the date as 20 September 2002?

[Figure]

P8, L9: On this page something went wrong with the line numbering. There is three times line number 5 marked. In "line 9" it should read: …….with differences of approximately 0.5 ppmv.

P8, L12-14: This is not clear. Why does than get more information in higher altitudes when the vertical resolution is higher?

P9, L5: What do you mean with active ozone? Do you mean assimilated ozone?

P8, section 4.3: SMILES has not been introduced yet.

P9, L6: I guess you mean "in" Section 4,.4?

P10, Figure 6 caption: It should rather read "We excluded the results of the 2002 Antarctic winter…….."

P10, L15-L16: Add years for the time periods the Rösevall and Sonkaew study were performed.

P11, Figure 7 caption: Same as above. Add the years.

P12, L28-30: This is not clear. I thought you used a vortex mean. In that case air from outside the vortex should have no influence.

P12, L15: This sentence should be rephrased so that it becomes clear what the authors want to say.

P14, L33: This sentence is a general explanation why ozone is destroyed, but does not explain why the largest ozone loss occurred in 2010/2011.

[Figure]

P15, L1: This sentence is misleading and should be rephrased.

P15, L2: What do you mean with NOx increase inducing ozone loss? Do you mean NOx driven ozone loss?

P15, L11: The reference to the study in preparation should be given.

P15, section 4.4.2., line 9: For which time period? Over the last years? Please clarify.

P16, L13: What is the Ap Index? What does it describe? How is this index derived?

P23, L25: The third class should be just called "intermediate" and all three classes should be put into quotation marks.

**Technical corrections:**
P1, L7: line break not correct
P1, L12: space between "Hemisphere" and "more" is missing.
P3, L11: Space between "Section" and "4.3" is missing.
P3, L16: It should rather read " ……..affected not only by chemical processes but also by transport.
P4, L6: Space between number and unit is missing.
P4, L23: "of" is obsolete.
P4, L24: space between measurements and full stop obsolete.
P6, Figure 2 title: It should read -83.1 S (not North!)
P7, Figure 3 caption: include "its" so that it reads "……gradient of potential vorticity

has its maximum. . . . . ."

P7, L4: It should read Hemisphere instead of Hemispheres and a comma after respectively should be added.

P9, last line: remove space between closing bracket and full stop.

P10, L16: remove space between closing bracket and full stop.

P11, L18 and 20: Add model after DIAMOND, so that it reads in both cases DIAMOND model.

P11, L20: remove space between closing bracket and full stop.

P11, L21: Please correct the sentence as follows: This discrepancy in ozone depletion is not only found for the 2004/2005 winter but also for the other Arctic winters.

P12, L34: Please correct the sentence as follows: This value is lower than in other studies not using. . . . . . ..

P12, L12: Please correct the sentence as follows: We have also compared ozone loss estimates for the Southern Hemisphere with the results from Kuttipurath et al. (2015). . . ...

P15, L4: It should read: . . . . . . .Arctic ozone loss using POAM (. . ...) and MIPAS (. . . . . .) measurements in 2002/2003. P15, Section 4.4.2, L8: Space between studies and citations is missing.

P15, Section 4.4.2, L18: either an ":" should be put between instruments and Environment satellite or the sentences should be connected by namely, the. . . . . . . . ..

P16, last line: rephrase as follows: . . . . . . ..presented by Fytterer et al. (2015) but supports their hypothesis.

P17, first line after Fig. 11: "the partial column of ozone and depletion" should rather read "the partial column of ozone and the column ozone loss".

Table 1 caption: Add "N" so that it reads 70 N.

P19, L3: Space between number and unit is missing.

P19, first line of the last paragraph: It should rather read: "In the Southern Hemisphere, the change in stratospheric ozone from year to year. . . . . .."

P21, last line: "altitudes" instead of height".

P22, Table 2 caption: It should read "ozone loss" rather than "ozone hole".
P22, Table 2 caption: Add "S" so that it reads 70 S.
P22, L22: It should read ". . . . . ., while being 0.3 ppmv lower than. . . . . .."
P23, L26: Add "ozone" so that it reads: "The maximum ozone depletion. . . . . . . .."

─────────────────────────────

---

## Referee Comment (RC2) · Anonymous Referee #2 · 28 Jul 2016

General comments

In their manuscript, Sagi and Murtagh present assimilated ODIN SMR observations to derive Arctic and Antarctic stratospheric chemical ozone loss for the period 2002 to 2013. Monitoring polar stratospheric ozone and attributing changes to chemistry and transport is an important topic within the scope of Atmos. Chem. Phys. By providing a decadal perspective using a sophisticated assimilation scheme this study significantly adds to what is known so far. I suggest publication in Atmos. Chem. Phys. after consideration of some comments given below.

The manuscript is generally well written, but contains a number of typos and slightly awkward phrases. I have indicated some suggested correction but in addition suggest

that the authors carefully double-check before submission of the revised manuscript.

The outline of the manuscript can be slightly improved in two ways: (a) the abstract is too long it should be focused on the new and most important findings. (b) I suggest to move the discussion of ozone depletion in subsections 4.4.1-4.4.3 into a new section 5.

Specific comments

p1, l14: Not fully clear why you primarily compare to Sonkaew et al.? Are SCIAMACHY data particularly useful here or is this one of a few studies that span many years?

p1, l16: "544GHz ozone loss" is jargon to be avoided. Better write "Ozone loss derived from the 544GHz measurements..." or similar.

p1, l17: which year does this refer to?

p1, l20 and following: You name three classes of chemical ozone loss in the Northern Hemisphere, but the following sentences do not clearly state what these three classes are. There is "cold type loss" in cold winters with maximum loss below 500K, "warm winter loss" at 700K, but the third class is not mentioned. Instead the discussion is on the SH. Please state this more clearly.

p2, l16: Winter 2010/11 was cold, but was it really "obviously different from other Arctic winters"?

p3, l3: suggestion: "Konopka et al. (2007) showed for the Northern Hemisphere winter 2002/2003 that above 600K ..."

p4, l1: This study uses ECMWF operational analyses. Can you briefly motivate why operational analyses rather than ERA-Interim re-analyses have been used here? For a multi-year study ERA-Interim may be more appropriate?

p.4, l2: "normally" -> "approximately" (?)

Figures 1 and 2: Wouldn't it be better to have the two channels shown for the same ozone profile? What exactly do the error bars for the ozone profile contain? Only the the statistical error?

p6, l32: The reference to Lait (1994) should be for modified PV, the reference for the vortex edge should be Nash et al. (1996).

p8, l11: Are the different averaging kernels between 544 and 501GHz considered in the assimilation process, in addition to the measurement precision?

p9, l7: The sentence "Thus, ..." does not refer to the previous sentences, as one may expect, but introduces the conclusion to use only 544GHz. This should be better stated explicitly.

p10, l15: It makes a lot of sense to consider the period 1 December to 14 March, but the argument that this is the best to compare with the other studies starting at 1 January does not make much sense.

p12, l5: "We do not find any clear problem due to the assimilation technique": Maybe you want to phrase more positively?

p15, l11: Another study that may be referred to here (and the discussion on p16) is Kiesewetter et al. (2010) who used assimilation of SBUV data to investigate inter annual variations of Arctic and Antarctic ozone in the mid- and lower stratosphere: Kiesewetter et al., A long-term stratospheric ozone data set from assimilation of satellite observations: High-latitude ozone anomalies, J. Geophys. Res., 115, D10307, doi:10.1029/2009JD013362.

p15, l4: "although similar": similar to what?

p16, l8: what exactly does it mean to "neglect the direct effect of solar radiation"?

p18: I suggest to better separate the discussion of chemical ozone loss and long-term changes due to a possible acceleration of the BDC.

Table 1: The maximum loss of -59DU by 28 March 2011 is much smaller than the inferred loss of almost 120 DU as given by Sinnhuber et al. (2011) or other studies. Can you comment?

p19, l6: an additional explanation may be that by inspection of Fig 8 the loss in 2002 occurs at low altitudes where the density is larger. Would be good if you could give a quantitative explanation (e.g. by providing partial column losses over different altitude regions). In particular for the following sentence "Another reason could be an error in the vertical descent..."!

p19, l8: This is the first time that heating rates from SLIMCAT are mentioned. Should of course be introduced in Section 2. (Reference needed!) The discussion of the uncertainties due to heating rate interpolation should be expanded and maybe moved to Section 2.

p20, l1: If the regression slope makes less sense, I suggest to remove and just use mean ozone as a reference. These details detract here from the main points.

p23, l7: The largest uncertainty is probably due to the still short time period?

Minor corrections

Throughout the manuscript the usage of "the" in many places does not follow standard practice. Sometimes "the" should be omitted, sometimes it is missing. I suggest that a native speaker carefully checks again for a revised version of the manuscript.

p1, l12: "Hemispheremore" -> "Hemisphere more"; "50% of the difference found": do you mean: "a difference of 50% found"? ; "parts per million volume" -> "parts per million by volume"

p1, l15 "using a SCanning..." > "using the SCanning..."

p3, l7: remove redundant phrase "This paper contains the following sections."

p3, l32: "quality of the transport is preserved": maybe better say "quality of the transport

is improved" or "tracer gradients are better preserved"?

p11, l25: Something wrong with the PV units. Typically 1 PV unit = 10(-6) K m2/kg/s, so Kelvin seems to be missing here?